

# Isolation, growth, and nitrogen fixation rates of the *Hemiaulus-Richelia* (diatom-cyanobacterium) symbiosis in culture

Amy E. Pyle[1], Allison M. Johnson[2] and Tracy A. Villareal[1]

[1] Department of Marine Science and Marine Science Institute, The University of Texas at Austin, Port Aransas, TX, USA
[2] St. Olaf College, Northfield, MN, USA

Corresponding author
Tracy A. Villareal,
tracyv@austin.utexas.edu

## ABSTRACT

Nitrogen fixers (diazotrophs) are often an important nitrogen source to phytoplankton nutrient budgets in N-limited marine environments. Diazotrophic symbioses between cyanobacteria and diatoms can dominate nitrogen-fixation regionally, particularly in major river plumes and in open ocean mesoscale blooms. This study reports the successful isolation and growth in monocultures of multiple strains of a diatom-cyanobacteria symbiosis from the Gulf of Mexico using a modified artificial seawater medium. We document the influence of light and nutrients on nitrogen fixation and growth rates of the host diatom *Hemiaulus hauckii* Grunow together with its diazotrophic endosymbiont *Richelia intracellularis* Schmidt, as well as less complete results on the *Hemiaulus membranaceus-R. intracellularis* symbiosis. The symbioses rates reported here are for the joint diatom-cyanobacteria unit. Symbiont diazotrophy was sufficient to support both the host diatom and cyanobacteria symbionts, and the entire symbiosis replicated and grew without added nitrogen. Maximum growth rates of multiple strains of *H. hauckii* symbioses in N-free medium with $N_2$ as the sole N source were 0.74–0.93 div d$^{-1}$. Growth rates followed light saturation kinetics in *H. hauckii* symbioses with a growth compensation light intensity ($E_C$) of 7–16 µmol m$^{-2}$s$^{-1}$ and saturation light level ($E_K$) of 84–110 µmol m$^{-2}$s$^{-1}$. Nitrogen fixation rates by the symbiont while within the host followed a diel pattern where rates increased from near-zero in the scotophase to a maximum 4–6 h into the photophase. At the onset of the scotophase, nitrogen-fixation rates declined over several hours to near-zero values. Nitrogen fixation also exhibited light saturation kinetics. Maximum $N_2$ fixation rates (84 fmol $N_2$ heterocyst$^{-1}$h$^{-1}$) in low light adapted cultures (50 µmol m$^{-2}$s$^-$1) were approximately 40–50% of rates (144–154 fmol $N_2$ heterocyst$^{-1}$h$^{-1}$) in high light (150 and 200 µmol m$^{-2}$s$^{-1}$) adapted cultures. Maximum laboratory $N_2$ fixation rates were ~6 to 8-fold higher than literature-derived field rates of the *H. hauckii* symbiosis. In contrast to published results on the *Rhizosolenia-Richelia* symbiosis, the *H. hauckii* symbiosis did not use nitrate when added, although ammonium was consumed by the *H. hauckii* symbiosis. Symbiont-free host cell cultures could not be established; however, a symbiont-free *H. hauckii* strain was isolated directly from the field and grown on a nitrate-based medium that would not support DDA growth. Our observations together with literature reports raise the

possibility that the asymbiotic *H. hauckii* are lines distinct from an obligately symbiotic *H. hauckii* line. While brief descriptions of successful culture isolation have been published, this report provides the first detailed description of the approaches, handling, and methodologies used for successful culture of this marine symbiosis. These techniques should permit a more widespread laboratory availability of these important marine symbioses.

# INTRODUCTION

The phytoplankton flora of the open sea is a diverse assemblage of prokaryotic and eukaryotic cells that span a size range of ~1 to 2,000+ µm in diameter. Nitrogen is often a limiting nutrient in the open sea, and planktonic nitrogen fixation (diazotrophy) occurs in tropical, subtropical systems and high latitude systems (*Zehr, 2011*; *Harding et al., 2018*). However, nitrogen fixation can occur in a wide variety of deep-sea and benthic habitats not traditionally associated with nitrogen-limitation (*Zehr & Capone, 2020*). Diazotrophy occurs only in prokaryotic cells, but a variety of symbiotic associations between diazotrophic prokaryotes and host eukaryotes are known (*Foster, Carpenter & Bergman, 2006*; *Foster & O'Mullan, 2008*; *Taylor, 1982*; *Villareal, 1992*; *Zehr & Capone, 2020*) and cover the range from obligate symbioses to loosely associated consorts (*Caputo, Nylander & Foster, 2019*; *Carpenter, 2002*; *Foster, Carpenter & Bergman, 2006*; *Foster & O'Mullan, 2008*). Of these, diatom-diazotroph associations (DDAs) are the most visible with records dating back to the early 20th century (*Karsten, 1905*).

Two types of marine diatom-cyanobacteria symbioses are known: diatoms in the genera *Neostreptotheca* and *Climacodium* that host coccoid cyanobacteria (*Carpenter & Janson, 2000*; *Hallegraeff & Jeffrey, 1984*), and diatoms that host filamentous, heterocyst-forming cyanobacteria of the genera *Richelia* and *Calothrix*. Little is known about the characteristics of the coccoid symbionts in diatoms, although the *Climacodium* symbiont is a diazotrophic *Crocosphaera* sp. (*Foster et al., 2011*). DDA symbioses involving heterocystous, diazotrophic cyanobacteria are abundant in both open ocean systems (*Dore et al., 2008*; *Villareal et al., 2011*; *Wilson et al., 2008*) and at intermediate salinities within the Amazon (*Foster et al., 2007*), Mekong (*Grosse et al., 2009*) and Congo River plumes (*Foster, Subramaniam & Zehr, 2009*). These marine regions differ greatly in their characteristics, suggesting either a great plasticity in physiological responses to environmental variables or undocumented differentiation within these symbioses. Symbiont integration with the hosts varies as well. In the *Rhizosolenia-Richelia* DDA symbiosis, the symbiont is located in the periplasmic space between the frustule and plasmalemma and has limited contact with the external environment (*Janson et al., 1999*; *Villareal, 1989*). The *Hemiaulus-Richelia* DDA symbiont is appressed to the nucleus and truly intracellular (*Caputo, Nylander & Foster, 2019*), consistent with its reduced

genome (*Hilton et al., 2013*). The *Chaetoceros-Calothrix* DDA symbiont is completely extracellular to the host diatom (*Foster, Goebel & Zehr, 2010*).

Despite their ubiquitous occurrence in tropical seas, the *Hemiaulus-Richelia* symbiosis was largely overlooked until epifluorescence microscopy revealed the cryptic *Richelia* symbiont (*Heinbokel, 1986*) and $N_2$ fixation was documented in individually picked chains of the symbiosis (*Villareal, 1991*). In addition to providing fixed N to the pelagic community, diatom-cyanobacteria symbioses play an important role in the nitrogen and carbon cycles of oceanic systems by virtue of their potential to sequester carbon to the deep sea via aggregation and sinking (*Karl et al., 2012*; *Subramaniam et al., 2008*). In the currency of oceanic nitrogen cycling, nitrogen derived from photosynthetic nitrogen fixation is generally balanced by a concurrent removal of atmospheric $CO_2$ (*Eppley & Peterson, 1979*). Thus, sinking material fueled by phototrophic diazotrophy represents a net removal of $CO_2$, and is a quantitatively important process in the transport of carbon to depth. DDAs, and particularly *Hemiaulus* symbioses, are of particular oceanographic significance. *Hemiaulus-Richelia* symbioses bloom at ~$10^3$ cells $L^{-1}$ frequently at the Hawai'i Ocean Time-series HOT (*Dore et al., 2008*; *Fong et al., 2008*; *Scharek et al., 1999*; *White, Spitz & Letelier, 2007*). At this location, they are the likely source of the summer export pulse that provides 20% of the annual carbon flux to 4,000 m in a 4–6-week window (*Karl et al., 2012*) and are regularly found on sinking particles (*Farnelid et al., 2019*). Subtropical front blooms at ~28–30°N in the Pacific (*Venrick, 1974*; *Villareal et al., 2012*; *Wilson, Villareal & Bogard, 2004*; *Wilson et al., 2008*) and in waters west and north of HI (*Brzezinski, Villareal & Lipschultz, 1998*; *Villareal et al., 2011*) suggest a basin scale significance. In the southwest Atlantic Ocean, *Hemiaulus hauckii-Richelia* blooms cover $10^5$+ $km^2$ and sequester 1.7 Tmol of carbon annually (*Carpenter et al., 1999*; *Subramaniam et al., 2008*) and $CO_2$ drawdown effects can extend to $10^6$ $km^2$ (*Cooley et al., 2007*). The large size, chain-formation, and tendency to aggregate (*Scharek et al., 1999*; *Villareal et al., 2011*) in the host *Hemiaulus* lead to an efficient export mechanism (*Yeung et al., 2012*) for both N and C.

Culture studies on the growth and physiological characteristics of these symbioses are limited. The external symbiont *Calothrix rhizosoleniae* has been cultured without its host (*Foster, Goebel & Zehr, 2010*) in both natural and artificial seawater medium. Cultures of the *Rhizosolenia-Richelia* symbiosis using amended seawater have been reported in the literature with growth rates up to 0.8 div $d^{-1}$ in fixed N-free medium (*Villareal, 1990*). In the *Rhizosolenia-Richelia* DDA, host and symbiont growth can be independent and symbiont-free host cells occur (but have reduced growth rates) even when no fixed N is present, possibly through use of N excreted by *Richelia* into the medium. Addition of nitrate rapidly results in the loss of symbionts as asymbiotic *Rhizosolenia* uses the added nitrate, increases its growth rate, and out-competes symbiotic *Rhizosolenia-Richelia* (*Villareal, 1989*, *1990*). Nitrogen fixation follows typical light saturation kinetics and can provide the entire N needs of the symbiosis (*Villareal, 1990*). Although oceanographically more significant than other *Rhizosolenia-Richelia* DDA (*Heinbokel, 1986*; *Subramaniam et al., 2008*; *Villareal, 1992*), there are no published culture-based data for the *Hemiaulus-Richelia* symbiosis.

Using nano-SIMS on field samples, *Foster et al. (2011)* were able to document the transport of recently fixed N from the symbiont *Richelia* to the host *Hemiaulus* in sufficient quantities to support growth; however, it is not known whether *Hemiaulus-Richelia* can grow exclusively on diazotrophically fixed N. Regardless, the symbiont is clearly advantageous to the host since, where examined, 80–100% of the *Hemiaulus* contain the symbiont (*Bar-Zeev et al., 2008*; *Heinbokel, 1986*; *Villareal, 1991*, *1994*) and 85–100% of the total phytoplankton N needs in the Amazon River plume can be met by *Hemiaulus* DDA diazotrophy (*Carpenter et al., 1999*; *Weber et al., 2017*). The symbiosis is not obligate for the host *Rhizosolenia* in DDA cultures (*Villareal, 1990*) and the field evidence suggests this may also be true for the host *Hemiaulus* (*Heinbokel, 1986*; *Kimor, Reid & Jordan, 1978*). This latter hypothesis has not been tested due to the difficulty in growing the *Hemiaulus-Richelia* host-symbiont pair in vitro.

In this article, we report the successful isolation of two species of the *Hemiaulus-Richelia* symbiosis into culture and expand on the brief culturing description reported in *Schouten et al. (2013)*. Using primarily *H. hauckii-Richelia* DDA strains, we document light-dependent growth rates, diel cycles of $N_2$ fixation, growth rate response to various forms of added nitrogen, and $N_2$ fixation rates. These parameters are essential to supporting modeling of DDA bloom formation and fate (*Follett et al., 2018*; *Stukel et al., 2014*). In addition, key differences between the *Hemiaulus* and *Rhizosolenia* DDAs are noted.

## METHODS AND MATERIALS

All culturing was conducted at the University of Texas Marine Science Institute (UTMSI) in Port Aransas, Texas. *Hemiaulus* strains containing symbionts were isolated by micropipette (*Andersen & Kawachi, 2005*) from the Port Aransas ship channel (27° 57′ 17.56″ N, 90° 03″ 00.48″ W) using material from either net tows (20–35 μm mesh nets, 1–3 min tows in the incoming tide) or whole water samples (incoming tide). The net tow sample was collected from a platform under a pier laboratory that both shaded the sample from direct sun the entire time as well as facilitating numerous short tows resulting in dilute samples. Both initial isolations and subsequent cultures of symbiont containing *Hemiaulus* were maintained in sterile filtered, fixed N-free YBCII media with L1 trace metal/EDTA additions (*Ohki, Zehr & Fujita, 1992*; *Chen, Zehr & Mellon, 1996*; *Guillard & Hargraves, 1993*), and final concentrations of 1 μm sodium glycerophosphate ($C_3H_7Na_2O_6P$), 2.6 μm sodium dihydrogen phosphate monohydrate ($NaH_2PO_4 \cdot H_2O$) and 35.7 μm sodium metasilicate ($Na_2SiO_3 \cdot 9H_2O$). Throughout the text, N-free or fixed N-free medium will refer to culture medium that has no added organic or inorganic N, recognizing that dissolved $N_2$ will be abundant as an N source for diazotrophs. Sterile filtered medium and seawater were generated using commercially available sterile tissue culture towers and reservoirs (0.22 μm pore size filters). Sterile filtration units were rinsed with ~50 ml of medium prior to use for culture medium. Both nylon and methyl cellulose 0.22 μm pore size filters were used with no apparent difference in results. All chemicals were reagent grade or better. The modified YBCII medium was checked with a hand-held refractometer before each use and adjusted to a salinity of 35 as

needed using 18 megaohm deionized water. Autoclaved tubes were rinsed with sterile filtered medium and then the tube filled with 15–20 mL of medium. This rinsing step was used for all flasks and tubes used for culturing.

Isolations were performed within 5–10 min of collection. Using a stereomicroscope, multiple *Hemiaulus* chains were rapidly isolated from the net tow material using hand-held borosilicate pipets drawn to a fine diameter in a gas flame. In our work, the drawn-out pipets were attached with tubing to a cotton-plugged mouthpiece (to prevent seawater aspiration) of fire-polished glass tubing. Mouth pipetting was used to carefully draw or expel the chain. If mouth pipetting is unacceptable, any form of fine control would provide adequate results. Multiple *Hemiaulus* chains were isolated into one well of a glass depression well plate (16 depressions) containing ~2 ml of medium per depression. Individual chains were then rinsed via serial transfer into other wells containing sterile medium. Extensive rinsing (5–6 rinses) of a single chain before isolating the next chain was much less successful than only 2–3 rinses before placing the chain into a tube of medium. When isolated directly into the N-deplete modified YBCII medium, contaminant growth was minimal even with only 1–2 rinses. These techniques resulted in symbiosis isolation free of other eukaryotes or cyanobacteria in ~30–40% of the attempts. Preliminary experiments used *H. hauckii* strain #9 isolated during Spring 2010. Subsequent *H. hauckii* experiments used isolates established during Fall 2010 (strain #22) and Fall 2011 (strain #83, #91, and #92). *Hemiaulus membranaceus* strain #82 was isolated in the Fall 2011. In all subsequent text, a strain designation indicates a culture of a host diatom containing one or more symbionts. While *Hemiaulus hauckii* strain #91 was used for most of the experiments, a single strain for the entire suite of experiments was not possible due to loss of the strain or the periodic loss of vitality noted in the results. Strains used are identified in the text and in Table S1.

The *Hemiaulus* DDA could be isolated for short-term growth into MET-44 (*Schöne & Schöne, 1982*) nutrient-amended (no added nitrate) sterile filtered seawater (0.22 μm filter equipped commercial sterile filtration units) collected at the isolation point. However, the *Hemiaulus-Richelia* symbioses required re-isolation from this MET-44 medium into the modified YBCII medium for successful maintenance >2–3 weeks. After isolation, cells were placed in a 25 °C incubator under cool white fluorescent illumination of 150–250 μmol m$^{-2}$ s$^{-1}$ on a 12:12 Light:Dark (L:D) cycle. All cultures were grown as batch cultures. Cultures had a high rate of sudden decline and death when kept in medium longer than 7–10 days and careful attention was required to transfer the cultures to new medium within this time frame. Experiments were initiated within 6 months of culture isolation; cultures failed to make auxospores and were eventually lost after approximately 1–2 years in culture. No attempt was made to culture axenically; bacteria were rarely visible in the cultures under phase contrast or differential interference contrast optics until senescence when cell mortality was substantial. *The H. hauckii* DDA was the primary experimental tool. *Hemiaulus membranaceus* DDA cultures were examined for general characteristics but were not the subject of intensive experimentation. In March 2017, *Hemiaulus* chains were observed in the Port Aransas ship channel from the Imaging Flow Cytobot data stream (*Campbell et al., 2010*, *2017*). Examination of net tow material noted

numerous asymbiotic *H. hauckii* chains and no symbiotic cells. Asymbiotic chains of *Hemiaulus hauckii* were isolated into N-replete (40 μm $NO_3^-$) MET-44 amended sterile filtered seawater as noted above. Unless otherwise noted, all experiments were conducted using modifies YBCII medium with no added nitrogen. Dissolved $N_2$ was the only available nitrogen source.

## Analytical methods

Cells were counted using a S52 Sedgewick-Rafter chamber on an Olympus BX51 epifluorescence microscope. Excitation/emission wavelengths for the epifluorescent filters used in counts and photography were 450 nm/680 nm (chlorophyll *a*), and 490 nm/565 nm (phycoerythrin). Both host cells and symbiont trichomes/heterocysts were enumerated. Percent symbiosis was calculated as the number of diatoms containing one or more *Richelia* trichomes divided by the total number of potential host cells. Growth rates (reported as div $d^{-1}$) were calculated using daily counts as the slope of the log of cell number over the change in time (*Guillard, 1973*) with the 95% confidence interval around the slope of the line calculated in Microsoft Excel.

Acetylene reduction assays (ARA) were performed as described in *Capone (1993)* corrected for ethylene solubility as described by *Breitbarth et al. (2004)* and assuming a mol ethylene reduced per mol $N_2$ conversion ratio of 4:1 (*Jensen & Cox, 1983* as modified by *Capone, 1993*). An SRI 8610C gas chromatograph (SRI Instruments, Torrance, CA, USA) equipped with a 30 cm silica gel column was used to quantify ethylene using a commercially prepared standard (GASCO Safeware Precision Gas Mixture, 10 and 100 ppm). Manufacturer-provided software (PeakSimple Chromatography Software) performed peak integrations. Standards were run prior to each day's run and at several points during the experiment. For each assay, 15 ml of culture sample was added to an acid-washed 25 ml incubation vial fitted with a gray chlorobutyl rubber serum stopper and crimped aluminum seals leaving 10 ml of headspace. Sterile-filtered medium was used as a control. A separate aliquot was retained for cell counts. One ml of acetylene generated from calcium carbide (*Capone, 1993*) was introduced, gently swirled for 15–30 s to equilibrate while minimizing contact between the serum stopper and the culture, then 100 μL of the vial headspace injected with a Hamilton gas-tight syringe and injected into the GC. Each injection required 5–7 min after an injection to return to baseline.

Chlorophyll *a* was determined on methanol-extracted (24 h, −20 °C) samples (10–25 ml aliquot) collected on 0.4 μm pore size polycarbonate filters using a non-acidification method (*Welschmeyer, 1994*). Initial tests indicated the filters used did not leach fluorescent compounds in the methanol. When chl *a* $cell^{-1}$ is referred to, it always includes both symbiont and host chl *a*. Sample fluorescence was read on a TD-700 Fluorometer (Turner Designs, Fresno, CA, USA).

For nutrients, a 25 mm, 0.22 μm pore-size membrane cellulose ester Millipore filter mounted on a syringe was rinsed with 5 ml of sample, filtrate discarded, and ten ml of sample medium was filtered and frozen. A SEAL Analytical QuAAtro autoanalyzer was used to determine dissolved inorganic phosphate (DIP), nitrate +nitrite (N + N), ammonium ($NH_4^+$), and silicate ($SiO_4^{-2}$) concentrations using the manufacturer's

recommended chemistries. The chemistries are similar to automated analyses published in *Grasshoff, Kremling & Ehrhardt (1999)* with changes in reagent concentration and wetting agents specific to the manifold chemistries. Detection limits were ~0.05 µm for N + N, $NH_4^+$ and P, and ~0.5 µm for Si.

## Growth-rate and $N_2$ fixation versus irradiance experiments

*Hemiaulus hauckii* symbiosis strains #9 and #91 were used for the irradiance-rate experiments. Initial experiments (Strain #9) used two light levels and are included for comparison. Detailed growth rates and $N_2$ fixation rates were measured in separate experiments using 7–8 different light levels (photosynthetic photon flux density) ranging from 15 to 600 µmol $m^{-2}$ $s^{-1}$ measured by a QSP-170B irradiance meter (Biospherical Instruments; Table S1). For growth rates, cultures were grown at the 7 experimental light levels for 7 days and remained at the assigned light level through the duration of the experiments. Symbiosis growth is used throughout this paper to refer to increases in host diatom numbers containing at least one symbiont. For $N_2$ fixation, strains were adapted to either 50, 150, or 200 (high light HL) µmol $m^{-2}$ $s^{-1}$ at 25 °C and a salinity of 35 under cool white fluorescent lighting for 7 days prior to the acetylene reduction assay. Each adaptation level was then exposed to 7–8 light levels for acetylene reduction assay.

## Diel pattern of $N_2$ fixation

*Hemiaulus hauckii* strains #22 and #92, and *H. membranaceus* strain #82 were used for the diel study (12:12 L:D cycles at 200 µmol $m^{-2}s^{-1}$) examining the daily rhythm of $N_2$ fixation on culture medium with no added N. Initial experiments on *H. hauckii* strain #22 utilized a set of 6 discrete time points between 06:00 and 21:00. Each incubation lasted 4 h with initial and final measurements taken in triplicate. Rates were normalized to heterocysts and used the center point of the 4 h incubation period as the time stamp. Subsequent experiments on *H. hauckii* strain #92 and *H. membranaceus* strain #82 utilized a high frequency time series approach in order to resolve changes occurring on an hourly basis or less. This approach used a series of individual measurements taken from a single vial over a period of up to ~12 h and was utilized for two reasons. First, individual assays injections required 5–7 min to return to baseline. Triplicate measurements therefore required 15–21 min during which ethylene production was occurring at measurable rates, could not be consideration true replication of the ethylene measurement. Averages of these triplicates would be unable to resolve rate changes on short time scales. The second reason for this approach was to minimize handling, agitation, and light/temperature variation of the samples. Six (*H. hauckii*) or 8 (*H. membranaceus*) paired vials were started at various time points in the diel cycle to permit overlap. Individual time series can be identified from the labeling in Table S1. Vials were sampled sequentially (1a, 1b, 2a, 2b, 3a, 3b, then repeated) yielding approximately 1–1.5 h between successive sampling of a single vial. The difference between successive measurements (ethylene per heterocyst) was normalized to the time difference between the two successive points (~1 to 1.5 h) and expressed as a rate (ethylene $heterocyst^{-1}$ $time^{-1}$). Eighty-nine (*H. hauckii)* and 78 (*H. membranaceus)* separate measurements were plotted against time using a 5-point

running average (center point plus two on either side) to smooth the data. Rates from different vial series overlapped in time, thus the 5-point average has rates from independent time series. Standard deviation was calculated on this 5-point series recognizing this is not a statistically useful value but only a metric for the noise in the data. Experimental cultures were adapted to at 25 °C under 200 µmol m$^{-2}$ s$^{-1}$ illumination (cool-white fluorescence bulbs) on 12:12 LD cycle. Experimental vials were incubated under these same conditions. Samples during the scotophase were collected/returned to the incubator in a darkened container and shielded from the dimmed laboratory lights during the assay.

### Nutrient addition experiments

Nitrogen source experiments addressed the effect of various inorganic N sources on symbiosis growth and N$_2$ fixation. In these experiments, *H. hauckii* strain # 83 was transferred to three 2 L autoclaved glass Erlenmeyer flasks containing the maintenance medium listed above amended with one of the following nitrogen sources: no added nitrogen (control), added nitrate (40 µm) or added ammonium (10 µm). Samples were maintained at 25 °C and a salinity of 35. Reduced ammonium concentrations were used to avoid toxicity effects; the nitrate concentration duplicated work on the *Rhizosolenia-Richelia* symbiosis (*Villareal, 1989*). Nutrient concentrations and cell abundance were sampled 10 times throughout the duration of the 20-day experiment. Nutrient analyses and cell counts were done in duplicate.

### Curve-fitting and statistics

Light-dependent growth was fit to the Jassby-Platt hyperbolic tangent function (*Jassby & Platt, 1976*) with a y-intercept term to permit calculation of compensation light intensity. The y-intercept term was omitted for the N$_2$ fixation rates vs. irradiance curves due to time-dependent decline in dark N$_2$ fixation that became evident in the diel measurements. When not omitted, the time-dependent decline in dark N$_2$ fixation noted in the diel experiment at the beginning of the scotophase resulted in a highly variable initial slope as well as a significant y-intercept (dark fixation rate) that was not consistent with the longer term rates after several hours in darkness. Delta Graph (Red Rocks Software, Las Vegas, NV, USA) was used for graphics as well as curve fitting of the growth and N$_2$-irradiance curves. *T*-tests were performed using the data analysis package in Microsoft Excel. Confidence intervals or standard deviations (noted in text) were calculated using Microsoft Excel software. Data from all figures are found in Table S1.

### RESULTS

*Hemiaulus hauckii* and *Hemiaulus membranaceus* with their symbiont *Richelia intracellularis* were successfully isolated multiple times. We found it was essential to remove the *Hemiaulus* from the net tow sample as quickly as possible (3–5 min after completion of the tow). Successful culturing resulted in rapidly growing chains of *Hemiaulus* reaching over 80 cells in length (Fig. 1). Multiple symbionts (usually 1–2, but never more than 4) were evident in the cells. Cultures were sensitive to handling, and

PeerJ ______________________________

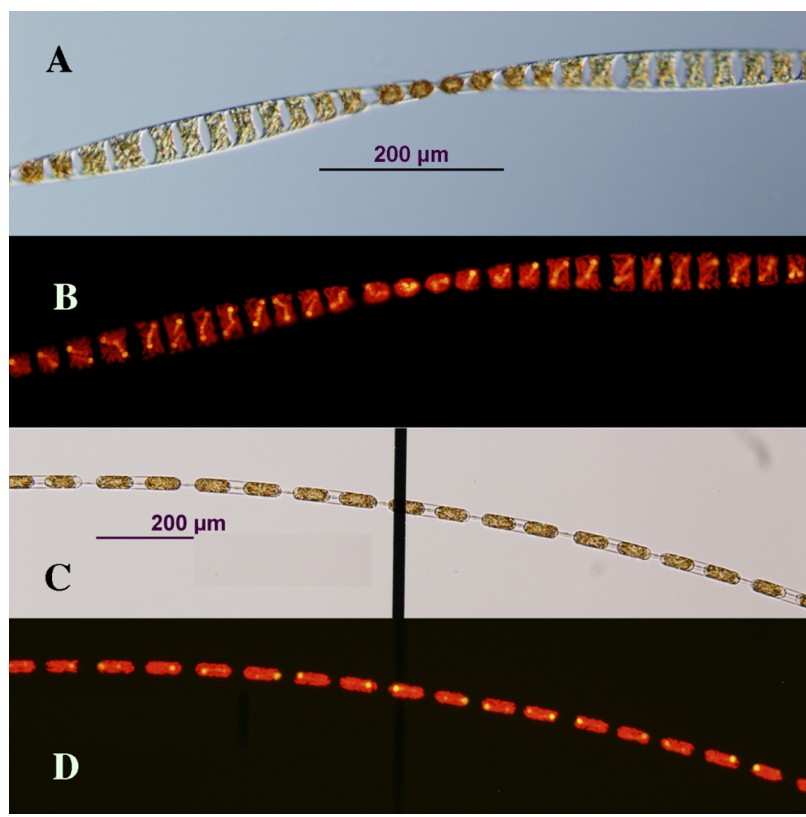

**Figure 1 Photomicrographs of *Hemiaulus membranaceus* (A and B) and *H. hauckii* (C and D) symbioses cultured in this study.** Images are paired photomicrographs of transmitted light micrographs (A and C) and light micrographs under epifluorescence (see "Methods") (B and D), and were taken from samples in a Sedgewick Rafter counting cell to minimize breakage of long chains. The vertical black bar in (C) is a marking from the Sedgewick Rafter cell. It lies under the chain; hence. it does not obscure details illuminated by epifluorescence in (D). Scale bar = 200 μm for all images. Image credit: A.E. Pyle.

swirling tubes to re-suspend chains resulted in chain breakage and decreased growth rates. Growth in undisturbed large volume containers (10 L+) resulted in complex aggregate formation. Strains were difficult to ship, and only one attempt out of approximately 15 resulted in successful establishment in another facility. A single auxospore-like structure was observed, but no cell diameter increases were observed in any of the cultures.

*Hemiaulus hauckii* strains used in this study ranged from 12 to 17.5 μm (up to 30 μm observed) in diameter (pervalvar axis presented in broad girdle view) with a total cell volume range of 7,012–23,574 μm$^3$. *H. membranaceus* cells were not measured. Since auxosporulation did not occur, the strains gradually decreased in diameter over a period of 1–2 years and eventually died out. Individual strains exhibited periods (weeks/months) of healthy growth (0.5–0.9 div d$^{-1}$) with little care required. This growth pattern was interspersed with intervals (days/weeks) of low growth rates that required substantial attention and multiple backups to prevent loss of the culture. These cyclic patterns were not linked to batches of culture medium or glassware. While not enumerated, bacteria were

rarely evident in light microscopy but certainly present since the cultures were not axenic. Reasons for the observed growth pattern variability remain unknown.

Individual symbiosis strains were routinely maintained in modified YBC-II medium with no added nitrogen. High densities of *Hemiaulus* and its symbionts were possible with no N added to the synthetic seawater medium (residual combined inorganic N < 0.1 μm). Maximum cell counts of the host *H. hauckii* reached ~10,000 cells mL$^{-1}$ with a maximum chl *a* concentration of 71 μg L$^{-1}$. Typical cell and chl *a* dynamics are shown in Fig. 2. High light (200 μmol m$^{-2}$ s$^{-1}$) chl *a* concentration reached a maximum approximately 3.5 times greater than the low light (50 μmol m$^{-2}$ s$^{-1}$) concentrations, although chl *a* per symbiosis (combined host and symbiont; multiple strains) remained approximately equal over time. In both light conditions, chl *a* per symbiosis was maximal (~4 to 5 pg chl *a* symbiosis$^{-1}$) in early exponential growth and declined over time to ~2 to 3 pg chl *a* symbiosis$^{-1}$. Extensive chain formation resulted in a high degree of variation in measurements.

Growth rates of *H. hauckii* in N-deplete medium (Fig. 3) followed light saturation kinetics with host and symbiont growth rates highly correlated ($r^2 = 0.98$, $p = 0.05$, *t*-test). Photoinhibition was not observed at the maximum light level used (500 μmol m$^{-2}$ s$^{-1}$). A modified Jassby-Platt curve fit (Article S1) yielded a realized maximum growth rate μ of 0.74–0.93 div d$^{-1}$ in replicated experiments (Fig. 3; Table 1). Light-saturated growth occurred with light saturation ($E_k$) occurring at 84–110 μmol m$^{-2}$ s$^{-1}$ and an initial slope (α) of 0.009 div d$^{-1}$(μmol m$^{-2}$ s$^{-1}$)$^{-1}$ in both irradiance curves. Compensation light intensity ($E_c$) calculated from the y-intercept and α varied from 7 to 16 μmol m$^{-2}$ s$^{-1}$.

Nitrogen fixation rates estimated by acetylene reduction were tightly linked to the light: dark cycle (Fig. 4). The 5-point running average was necessary to smooth the variable point-to-point time series rates into a general diel curve. Two separate experimental treatments (the 4-h incubations and the 5-point averaging series) indicated the maximum acetylene reduction rate in both *H. hauckii* and *H. membranaceus* DDA occurred approximately 4 h into the photophase (12:12 photoperiod) with a broader maximum acetylene reduction rate extending for 4–6 h. Acetylene reduction declined over several hours at photophase end to low (1–10% maximum values) but still measurable rates during the scotophase in both *H. hauckii* (Fig. 4A) and *H. membranaceus* (Fig. 4B) DDAs. Unlike the 4-h discrete incubation diurnal pattern seen in Strain #22, *H. hauckii* strain #92 rates maintained high values until the end of the photophase (Fig. 4A). *Hemiaulus membranaceus* DDA rates were more symmetrically distributed around the middle of the photoperiod (Fig. 4B). In both data sets, the rates reached a maximum in the range of 45–55 fmol N$_2$ heterocyst$^{-1}$ h$^{-1}$.

Nitrogen fixation-irradiance rates followed a light saturation curve (Fig. 5) fit to the hyperbolic tangent function. At the 150 and 200 μmol m$^{-2}$ s$^{-1}$ adaptation level ($r^2$=0.95 and 0.97, respectively), the curve-fit maximum N$_2$-fixation rates was 155 and 144 fmol N$_2$ heterocyst$^{-1}$ h$^{-1}$, respectively. The maximum rates (light-saturated) at 150 and 200 μmol m$^{-2}$ s$^{-1}$ adaptation level were significantly ($p < 0.01$, *t*-test) greater than the maximum (light-saturated) rate (86 fmol N$_2$ heterocyst$^{-1}$ h$^{-1)}$ noted in cultures adapted to 50 μmol m$^{-2}$ s$^{-1}$. The initial slope (light limited portion) of the N$_2$ fixation curve was

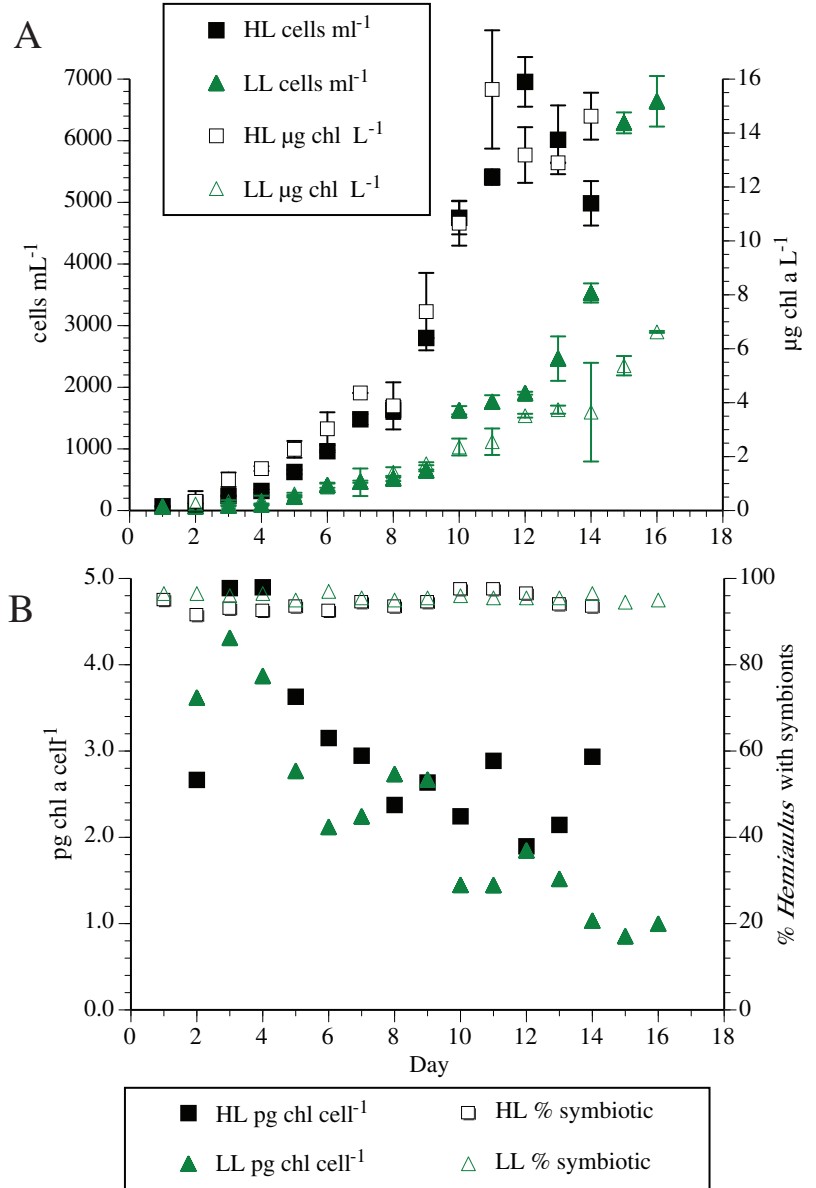

**Figure 2 Typical *Hemiaulus hauckii* growth curve at two irradiance levels in modified YBC-II medium with no added nitrogen using strain #91.** (A) Cell abundance and bulk chlorophyl *a* concentration. (B) Chlorophyll *a* content per host cell (sum of host and symbiont chl *a*)  and symbiont presence in hosts. High light (HL: 200 $\mu$mol m$^{-2}$ s$^{-1}$); Low light (LL: 50 $\mu$mol m$^{-2}$ s$^{-1}$). Values are means of duplicates ± standard deviation.   

approximately 75% higher in the 50 $\mu$mol m$^{-2}$ s$^{-1}$ adapted culture than the 150 and 200 $\mu$mol m$^{-2}$ s$^{-1}$ adaptation level.

Preliminary experiments in 2010 found that *H. hauckii* strain #9 did not utilize nitrate (Table S2). Subsequent replication experiments found that 40 $\mu$m nitrate was not used by a different *H. hauckii* symbiosis strain (#83) in experiments conducted 1 year later (Fig. 6). Ten $\mu$M added ammonium declined to ~0.4 $\mu$m in 13 days and then remained constant thereafter (Fig. 6). *Hemiaulus hauckii* strain #83 drew down P and Si under all the

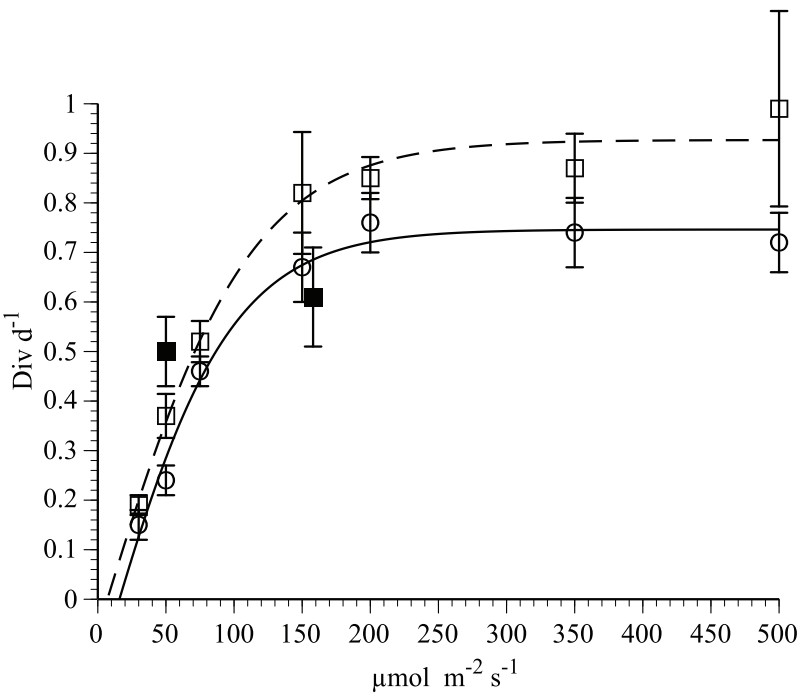

**Figure 3 Irradiance-growth rate relationships for *Hemiaulus hauckii* symbiosis strains #91 and #9.** Open circles and open squares are from strain #91 (7 points) in two separate experiments, solid squares (2 points) are from strain #9 measured approximately 1 year prior to strain #91. Error bars are 95% confidence intervals. Data from strain #91 are fit to a hyperbolic tangent function. Curve fit parameters are listed in Table 1 and in Article S1.

**Table 1 Results from the modified hyperbolic tangent function growth rate-irradiance and hyperbolic tangent function $N_2$-fixation-irradiance curve fit.** The three $N_2$-fixation experiments were adapted to the given light levels for 7 days. The growth rate experiments were adapted at each of the light levels for 7 days. Strain 91 was used in these experiments.

| Measurement | Incubation $E^1$ | Initial slope $(\alpha)$ | Realized maximum rate | $E_c^1$ | $E_k^1$ | $R^2$ |
|---|---|---|---|---|---|---|
| $N_2$ fixation | 50 | 2.079[2] | 84 | – | 41 | 0.75 |
| $N_2$ fixation | 150 | 0.905[2] | 155[4] | – | 170 | 0.95 |
| $N_2$ fixation | 200 | 1.197[2] | 144[4] | – | 120 | 0.97 |
| Growth Rate | Light gradient | 0.009[3] | 0.76[5] | 15 | 84 | 0.99 |
| Growth Rate | Light gradient | 0.009[3] | 0.93[5] | 7 | 110 | 0.99 |

**Notes:**
[1] $\mu mol\ m^{-2}\ s^{-1}$.
[2] $(fmol\ N\ heterocyst^{-1}\ h^{-1})\ (\mu mol\ m^{-2}\ s^{-1})^{-1}$.
[3] $(div\ d^{-1})\ (\mu mol\ m^{-2}\ s^{-1})^{-1}$.
[4] $(fmol\ N\ heterocyst^{-1}\ h^{-1})$.
[5] $div\ d^{-1}$.

available N sources at approximately equal rates. The addition of ammonium in an experimental comparison resulted in higher percentages (up to 48%) of asymbiotic cells in exponential growth than when either nitrate (10–20%) was added or no N was present in the medium (10–20%) but the strain was not grown free of its symbiont (Fig. 6).

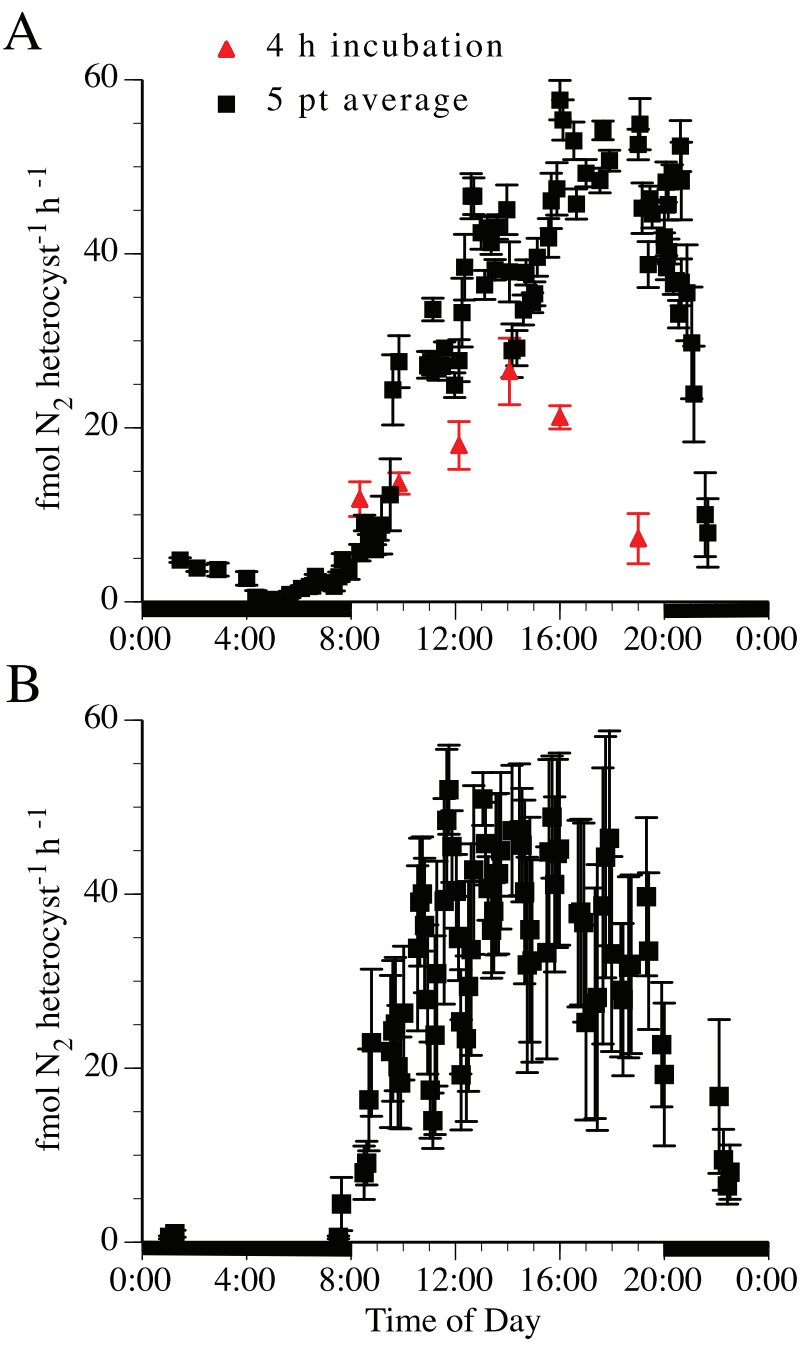

**Figure 4 Diel patterns of N$_2$ fixation in N-free medium.** (A) *Hemiaulus hauckii* symbiosis (Strain #92 = black symbols; Strain #22 = red symbol). (B) *H. membranaceus* symbiosis (Strain #82). Growth rates (*H. hauckii symbiosis*) were 0.35 ± 0.05 div d$^{-1}$ (strain #91) and 0.43 ± 0.10 div d$^{-1}$ (strain #22). Growth rate of the *H. membranaceus* symbiosis was 0.56 ± 0.10 div d$^{-1}$. Dark bars indicate nighttime. Both cultures were grown at 200 μmol m$^{-2}$ s$^{-1}$. See text for details of the methodology for the 4 h and 5 pt average measurements. Error bars are standard deviation.

A symbiont-free strain of *H. hauckii* was maintained from March 2017 to August 2017 on a solely nitrate enriched, natural seawater medium (MET-44). Ammonium concentrations in the aged stock seawater were 0.5 μm or less. When isolated and growing,

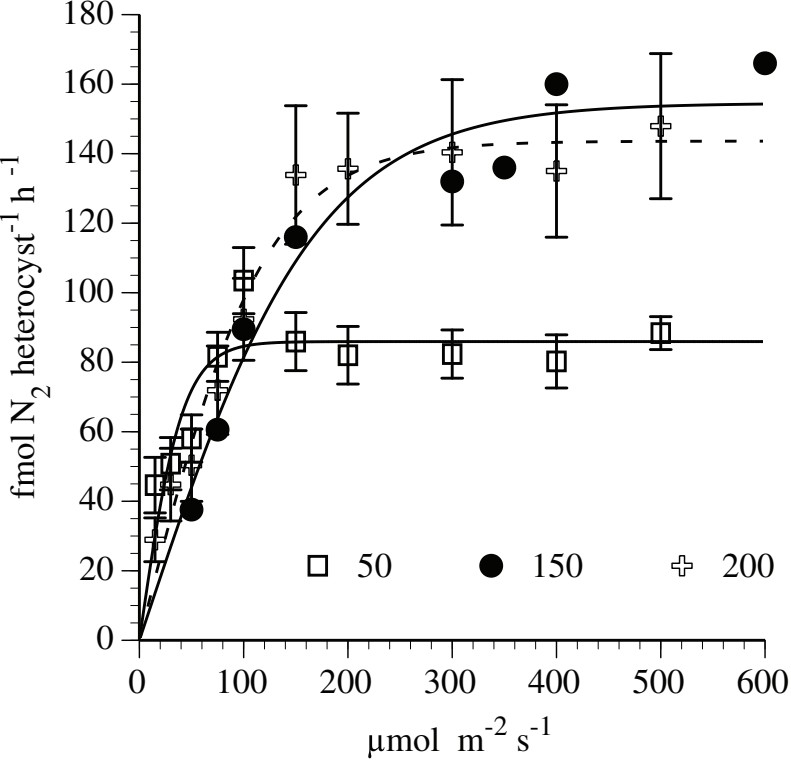

**Figure 5 Irradiance-N$_2$ fixation rate relationships for the *Hemiaulus hauckii* symbiosis (Strain #91) adapted to three light intensities.** Error bars are 95% confidence intervals.

it was confirmed in April 2017 to be symbiont-free by epifluorescence microscopy and maintained in a seawater-based culture medium (MET-44) that would not support the DDA strains. The strain was lost during Hurricane Harvey in August 2017 and no further information was collected.

## DISCUSSION

Physiology and rate measurements of *Hemiaulus* symbioses have previously been limited to field collected and incubated samples. Using a modification of an artificial seawater medium, we have successfully and reproducibly cultured two species of *Hemiaulus* with their symbiont. *Caputo, Nylander & Foster (2019)* also reported brief success using an artificial medium; *Hilton et al. (2013)* reported genetic sequences from *Richelia* extracted from *Hemiaulus* grown using these methods.

Greatest isolation success was found when the cells were rapidly removed from the net tow cod-end, suggesting sensitivity to the various exudates found in these concentrated samples. In addition, the seawater was sterile filtered rather than autoclaved or pasteurized. Sterile filtration leaves the carbonate system and medium pH unaltered compared to heat treatment; however, viruses are not inactivated. Little is known of virus/DDA interactions, but viruses play a significant role in diatom mortality in general (*Kranzler et al., 2019*) and could be a problem for stable cultures.

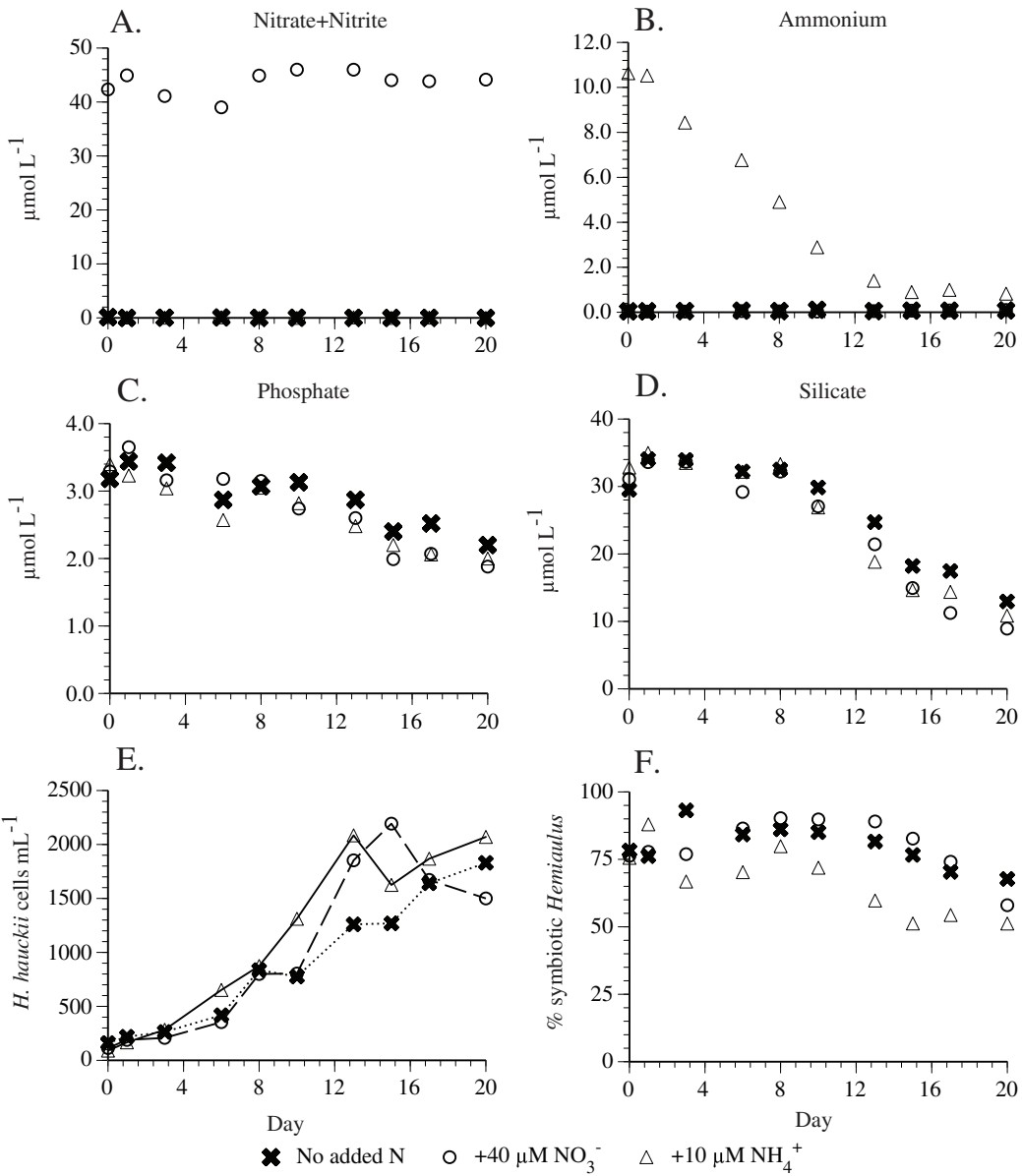

**Figure 6 *Hemiaulus hauckii* symbiosis growth in modified YBC-II medium under 3 nitrogen sources (N$_2$, nitrate, ammonium; Strain #83).** (A) Nitrate concentrations with and without added nitrate. (B) Ammonium concentrations with and without added ammonium. (C) Phosphate concentrations in the three nutrient conditions. (D) Silicate concentrations in the three nutrient conditions. (E) Cell abundance in the three nutrient conditions. (F) %*Hemiaulus* with symbionts in the three nutrient conditions. Only two symbols are visible in (A) and (B) due to overlapping near-zero values.

In addition, culture media designed to support phytoplankton may not support essential phycosphere components or may support difficult to remove lethal bacteria (see *Van Tol, Amin & Armbrust, 2017* for an example). The inability to culture *Hemiaulus* in a seawater-based enrichment medium used for concurrent *Rhizosolenia-Richelia* cultures suggests that the additional trace metal and chelation in our modified YBCII medium was required for sustained growth or that water quality issues are critical.

While we did not perform systematic comparisons, seawater from the Port Aransas pass is heavily influenced by both the inshore bays and coastal Gulf of Mexico. Our modified YBCII medium is free of these influences and we speculate provides a more consistent chemical environment. These difference highlights differing growth needs, sensitivities or tolerances of the *Hemiaulus* and *Rhizosolenia* DDAs that remain to be described. The oscillation between rapidly growing, apparently healthy cultures and less vigorous cultures is clearly an impediment to sustained culture as is the lack of auxospore formation. None of the isolations persisted for more than ~3 years making detailed work on model strains problematic at this time.

Previous estimates of $N_2$ fixation tracked $^{15}N$ isotope movement from the *Richelia* symbiont heterocysts to the host *Hemiaulus* cells using single-cell methods (*Foster et al., 2011*) and estimated that it was sufficient to support cell growth with a turnover time of up to 0.59 div d$^{-1}$. *Foster et al. (2011)* rate measurements for *H. hauckii-Richelia* ($n = 17$) averaged 20.4 ± 18.5 (std. dev.) fmol N heterocyst$^{-1}$ h$^{-1}$ (range 1.15–50.4 fmol heterocyst$^{-1}$ h$^{-1}$ ). These heterocyst normalized rates (Foster et al.'s Table 1 footnote), are lower than the rates observed in the cultures (up to 155 fmol $N_2$ heterocyst$^{-1}$ h$^{-1}$). Our culture growth rates (non-limiting conditions) suggest some degree of limitation in their field collections. Light/growth rate adaptation to 150–200 µmol m$^{-2}$ s$^{-1}$ PAR was concurrent with a maximum $N_2$ fixation rate approximately 6 times higher than the maximum rates observed by *Foster et al. (2011)*.

Data digitized (Plot Digitizer from SourceForge, Slashldot Media, 225 W. Broadway, Suite 1600, San Diego, CA, USA: https://sourceforge.net/) from *Carpenter et al. (1999)* Fig. 2, allows comparison of our symbiosis culture $N_2$ fixation rates to those from an Amazon River plume bloom where *Hemiaulus* DDA abundance reached $1.6 \times 10^6$ heterocysts L$^{-1}$. After extracting their $N_2$ fixation rates (as mg N m$^{-2}$ d$^{-1}$) and concurrent heterocyst abundance from *Carpenter et al. (1999)* Fig. 2, we determined that their rates ranged from 0.6 to 40.3 fmol N heterocyst$^{-1}$ h$^{-1}$ and were ~8 fold lower than the maximum rates seen in cultures (note the unit conversion and comparison: mg N, fmol N or fmol $N_2$ fixed). These rates were generated from material collected by net and then prescreened to remove *Trichodesmium*. Based on our isolation attempts, this handling probably adversely affected the rate. *Carpenter et al. (1999)* also reported undetectable nitrate uptake in the *Hemiaulus* DDA bloom, a result consistent with our culture observations that these DDA strains did not utilize nitrate.

Many growth characteristics of *H. hauckii-R. intracellularis* are similar to the *Rhizosolenia clevei-R. intracellularis* symbiosis. Maximum growth rates are slightly less than 1 div d$^{-1}$ and are similar between the two DDAs despite their significant size difference. Growth rates are not photoinhibited up to 500 µmol photons m$^{-2}$ s$^{-1}$. Rapidly growing cells form extensive chains. Culture agitation, albeit qualitatively measured, negatively affects chain formation and possibly growth rates. The diel pattern of nitrogen fixation in the *Hemiaulus* DDA cultures parallels the diel nifH nitrogenase gene expression seen in field samples of both *Hemiaulus* DDAs (*Zehr et al., 2007*), the *Rhizosolenia* DDA (*Harke et al., 2019*) and both gene expression and acetylene reduction in the *Calothrix* symbiont of the *Chaetoceros* DDA (*Foster, Goebel & Zehr, 2010*).

The differential nitrate use by the *Hemiaulus* and *Rhizosolenia* DDAs is a significant difference between the two DDAs. Preferential $NO_3^-$ utilization drove a higher host growth rate in a strain of the *Rhizosolenia* DDA, eventually leading to symbiont-free host cultures (*Villareal, 1990*) growing solely on $NO_3^-$. In field studies where $N_2$ appeared to be the primary N source, the *Rhizosolenia* host and symbiont DDA were tightly coupled (*Harke et al., 2019*). There are at least two mechanisms that could produce this result in the *Rhizosolenia* DDA: downregulation of symbiont diazotrophy by exposure to $NO_3^-$ due to its extra-plasmalemma location and/or induction of host nitrate reductase pathways. The latter would result in diminished carbon flow to the symbiont in order to support nitrate assimilation into protein. Neither of these mechanisms appear to have occurred in the *H. hauckii* DDA strains we used. These results were replicated in individual experiments 4 years apart on different strains, excluding the possibility that the results were a laboratory condition artifact. For the *Hemiaulus* DDA, either nitrate cannot be used or diazotrophic supply exceeded any immediate N demand by the symbiosis and suppressed $NO_3^-$ uptake. In contrast, ammonium was used and resulted in elevated percentages of symbiont-free hosts, but not a symbiont-free culture. The free-living marine cyanobacterium *Trichodesmium* can use $NO_3^-$ either preferentially or concurrently during diazotrophy as an N source (*Holl & Montoya, 2005*; *Klawonn et al., 2020*; *Mulholland & Capone, 2000*) and other diazotrophs can simultaneously use $N_2$ and $NO_3^-$ (*Inomura et al., 2018*). It is unusual for $NO_3^-$ not to be used at all due to the higher overall energetic cost of nitrogen fixation added to the costs maintaining specialized cellular structures in diazotrophs (*Inomura et al., 2018*). However, in the UCYN-A/haptophyte symbiosis, the host haptophyte only assimilates diazotrophically fixed $N_2$ even in the presence of combined DIN (*Mills et al., 2020*). Thus, while unusual, the *H. hauckii-Richelia* symbiosis is not unique. The truly intracellular location of the *Hemiaulus* symbiont (*Caputo, Nylander & Foster, 2019*) clearly limits its contact with the environment and the potential impact of $NO_3^-$, but our observations also require the host *Hemiaulus* to be unresponsive to external nitrate.

In contrast, *Hemiaulus* spp. (no information on symbionts) has been reported with growth rates up to 2.2 div $d^{-1}$ (*Furnas, 1991*) in field experiments and 3.8 div $d^{-1}$ in nitrate-based laboratory medium (*Brand & Guillard, 1981*). While the symbiont presence is undocumented but seems unlikely given the DDA growth rates reported in our paper of <1.0 div $d^{-1}$ as well as by modeled symbiont diazotrophy (*Inomura et al., 2020*). The *Furnas (1991)* and *Brand & Guillard (1981)* reports, as well as our briefly established asymbiotic strain on medium that would not support *Hemiaulus* DDA growth, all suggest that symbiont-free strains of *Hemiaulus* are extant in the modern ocean. *Hemiaulus* DDAs had an ancestral origin 50–100 million years ago (*Caputo, Nylander & Foster, 2019*), but asymbiotic *H. hauckii* strains apparently still persist in the modern ocean. We suggest this data supports, but does not prove, that the *Hemiaulus* DDA, with its close metabolic coupling of the host-symbiont nitrogen metabolism (*Foster & Zehr, 2019*; *Hilton et al., 2013*), is obligate and that symbiotic host *Hemiaulus* spp. are distinct from asymbiotic *Hemiaulus* strains. These asymbiotic strains should provide an invaluable tool for examining evolutionary processes in DDAs.

The growth rate and $N_2$ fixation results provide useful input to models examining the biogeochemical impact of the *Hemiaulus* DDA blooms in oceanic regions. The Amazon River plume is particularly noteworthy in that it has an explicit model describing the ecological-biogeochemical impacts. *Stukel et al. (2014)* model incorporated high generic $N_2$-based DDA growth rates >1 div d$^{-1}$ with asymbiotic cells growing on ambient N at somewhat greater rates. Our experimental results are much lower for growth on $N_2$ (maximum ~0.9 div d$^{-1}$) and indicated no nitrate use. Non-diazotrophic, asymbiotic *Hemiaulus* growth rates from the literature are much higher than $N_2$-based DDA rates. These are significant alterations in the input values available to *Stukel et al. (2014)*.

In addition, our results for *H. hauckii* DDAs found no evidence of growth rate photoinhibition at the highest light level used (500 µmol m$^{-2}$ s$^{-1}$). While instantaneous solar PAR may reach ~2,000 µmol m$^{-2}$ s$^{-1}$ (*Björkman et al., 2015*) at Station ALOHA near Hawaii (22° 45′ N 158° 00′ W), average daily PAR incident at Sta. ALOHA over the diurnal is ~850 µmol m$^{-2}$ s$^{-1}$ from June to August (calculated from *Letelier et al., 2017*). Vertical mixing rates will both reduce the time averaged PAR exposure exponentially with the depth of mixing as well as being rapid enough to preclude general phytoplankton photoacclimation (*Tomkins et al., 2020*). Thus, it seems possible that in-situ PAR values would not photoinhibit these DDA strains. However, damaging effects by solar UV wavelengths (*Zhu et al., 2020*) require further examination.

*Follett et al. (2018)* and *Inomura et al. (2020)* utilized *H. hauckii* DDA growth rates extracted from *Pyle (2011)* for modeling applications. Our report presents the full range of data in Pyle's work and notes that rates can be ~ 0.2 div d$^{-1}$ higher than the values used by *Follett et al. (2018)* depending on the strain used. These higher rates are consistent with the mechanistic model of *Inomura et al. (2020)* in that host carbon fixation is substantial enough support to the symbiont $N_2$ fixation rates required for the unit DDA growth. This host derived carbon is likely to also be the reductant and energy source required to support the lengthy decline of $N_2$ fixation rates at the beginning of the scotophase noted in the diel experiment (Fig. 4). Further experimental verification is required.

When comparing rates, the possibility of strain-specific variation between *Foster et al. (2011)* Pacific Ocean collections, *Carpenter et al. (1999)* field collections and our Gulf of Mexico isolations cannot be excluded. Symbionts of the 3 diatom host genera have diverged with strong host specificity within diatom host genera (*Foster & Zehr, 2006*; *Janson et al., 1999*). *Bar-Zeev et al. (2008)* noted evidence of seasonally varying *Hemiaulus*-DDA dominated *Richelia* clades in the Mediterranean but there is little data to assess how physiological characteristics vary with habitat. *Rhizosolenia* and *Hemiaulus* DDA symbionts appear limited to vertical transmission during division or possibly transmission during auxosporulation (*Foster & Zehr, 2019*) raising the possibility of genetic drift of various degrees within populations (*Bar-Zeev et al., 2008*).

## CONCLUSIONS

Two symbiotic associations between host diatoms and their intracellular heterocystous cyanobacterium (*Hemiaulus hauckii - Richelia intracellularis* and

*Hemiaulus membranaceus-Richelia intracellularis*) were successfully cultured for up to 3 years on artificial seawater medium. The $N_2$-fixation and growth rate data provided here are, to our knowledge, the first published laboratory-based data for the *Hemiaulus* DDA. This work provides details on isolation techniques that proved key to successful culturing. The symbioses are sensitive to handling, requiring rapid collection and isolation for successful growth. The cultures did not undergo sexual reproduction, and the lack of auxosporulation and concurrent size increase is a barrier to long-term stable culture. Both symbioses grow without added nitrogen other than dissolved $N_2$ and are supported at maximum growth rates solely by symbiont nitrogen fixation. Maximum growth rates of the intact diatom-cyanobacterium symbiosis are <1 div $d^{-1}$ and are similar to the reported rates for another diatom-cyanobacterium symbiosis (*Rhizosolenia clevei-Richelia intracellularis*). Unlike the *Rhizosolenia clevei-Richelia intracellularis* symbiosis, the *H. hauckii – Richelia intracellularis* symbiosis does not assimilate nitrate. Nitrogen fixation by the heterocystous symbiont while within the host diatom has a clear diel pattern with maximum rates occurring during the photophase. The culture nitrogen fixation rates are consistent with field measured rates; however, maximum culture rates are ~6 to 8 times previously measured field rates. Both growth and nitrogen fixation rates follow light saturation kinetics. These data provide direct input for parameterization of light-dependent growth and nitrogen fixation in biogeochemical models.

Both literature reports and our isolation of a nitrate-utilizing, symbiont-free *Hemiaulus* culture are consistent with distinct symbiont-free and symbiont-containing lines of the diatom *Hemiaulus*. If correct, these different lineages would be useful models for understanding the evolution of these symbioses in diatoms.

### Funding
This work was supported by the National Science Foundation grants OCE 0726726 and OCE 1923667. The funders had no role in study design, data collection and analysis, decision to publish, or preparation of the manuscript.

### Grant Disclosures
The following grant information was disclosed by the authors:
National Science Foundation: OCE 0726726 and OCE 1923667.

### Competing Interests
Amy Pyle is employed by Blanton & Associates Inc., Austin, Texas, USA. Allison Johnson (now Allison McMillen) is employed by Park Dental Blaine, Northfield, Minnesota, USA.

### Author Contributions
- Amy E. Pyle conceived and designed the experiments, performed the experiments, analyzed the data, prepared figures and/or tables, authored or reviewed drafts of the paper, and approved the final draft.

- Allison M. Johnson performed the experiments, analyzed the data, authored or reviewed drafts of the paper, and approved the final draft.
- Tracy A. Villareal conceived and designed the experiments, analyzed the data, prepared figures and/or tables, authored or reviewed drafts of the paper, and approved the final draft.

## Data Availability

Raw data used to create the figures and descriptive information on the curve fit selection are available as Supplemental Files.

## Supplemental Information

Supplemental information for this article can be found online at http://dx.doi.org/10.7717/peerj.10115#supplemental-information.

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
