# Peer review of "Isolation, growth, and nitrogen fixation rates of the Hemiaulus-Richelia (diatom-cyanobacterium) symbiosis in culture"

_PeerJ, doi:10.7717/peerj.10115_

## Round 0.1 · original submission · Major Revisions

Please provide a point-by-point response to all of the reviewers' comments along with your revised manuscript.

Reviewer 1 ·

Basic reporting

In general, the manuscript is written in clear, unambiguous, professional English. The intro and background show sufficient context and motivation for the experiments. This area of research is of great interest to the DDA community as there is very limited laboratory experimentation or published isolation methods for these important organisms. The figures were relevant although I note some comments regarding apparent inconsistencies (see line-by-line general comments). Raw data underlying individual figures was provided in a spreadsheet. It would be nice to know which strains were used with which raw data points, where applicable. I think this is mentioned in the text and figure legends but was hard to keep straight. Having it labeled in the figures and supplemental would help with clarity. Some data appears to be missing, for instance on Line 125 it is indicated that growth rates were measured at 7 different light levels, but much of this data is missing from the results and supplemental data. Perhaps this data was lost due to Hurricane Harvey?

Experimental design

The research conducted herein is very relevant to the field and provides much-needed information on culturing, growth, n-fixation rates, and ecology of DDA’s formed by Hemiaulus-Richelia. In general, the methods were well written and relevant to the goals of the study. In some cases, however, the methods were a little hard to follow as written and some suggestions are listed below in the General Comments. In particular, a statement to the reason different strains were used would help clarify this point to readers. Also, more clarity as to which strains were used in which figure would help. A more detailed description of statistical methods to infer significance is needed.

Validity of the findings

As mentioned elsewhere, all underlying data for the figures are provided, but not all underlying experimental results mentioned in the methods. These data would be of interest to the readers I feel, assuming they were not lost in the hurricane. The discussion of results and intercomparing with prior field and laboratory observations, as well as cross genera comparisons, were robust and very well written. I truly enjoyed reading this manuscript. The manuscript will have a large impact on the field and is very relevant to our ecological understanding of these keystone organisms.

Additional comments

Lines 12 to 13 – looks like a space was between citations
Line 35 – “Richealia” is spelled wrong
Line 56 – there is a “cc” after “medium”
Line 110 and 112 – it would be helpful to reviewers if a note is given as to why the use of different strains. In addition, rather than requiring the reader to go look at the figure legends or further in the text for environmental conditions of the experiments, it would be helpful if they were spelled out here as the N and light experiments are done. But maybe its just that sentence which is confusing.
Line 134 – add ‘nm’ after 565
Line 154 - italicize “a” in chlorophyll, here and elsewhere, also, how much volume for chl measurements?
Line 159 – what is the 10 ml sample medium used for? I assume the nuts, but since it follows the method it reads as though it will be used for a different analysis. consider moving up.
Line 160 – what stats were conducted with Excel? and on which data sets?
Line 191 – I think a period is supposed to be here after the volume range
Line 197 – Perhaps heterotrophic bacteria? Was any observations made on their numbers?
Line 205 – chlorophyll is now abbreviated where on line 204 it was not
Line 236 – what stat?
Line 242 – were these cultures axenic? If not, how might heterotrophic remineralization impact measured N+N values?
Figure 4 – How are the individual isolates represented in the figure? Are these the black squares? Not sure I see diamonds. If so, consider either using different colors to designate strain or additional shapes. If red squares are a 4 h incubation, how do the points span from 8am to 8pm?
Figure 6 – Where is the 3rd N source? Or is this the -N? No closed squares are visible. If strain #9 is here, it should be called out in the figure as well as mentioned in the methods for this experiment. Also, legend says only 1 um of NH4 was added whereas methods and results state 10um.

Reviewer 2 ·

Basic reporting

The language in this manuscript is clearly written. Take a closer look at the use of hyphens for compound adjectives. Because of the spotty use of some strains, communicating the experimental design sometimes became muddled. Comparisons to Rhizosolenia early in the manuscript led me to believe that there would be data presented from this strain, so I was left confused. The figures are clear, but in some cases require a legend or the captions need review (see specific comments below). Data are provided and I was hoping it could clarify questions I had regarding the N2 fixation calculations, but I now have more questions.

Experimental design

Isolating diatom-diazotroph associations from the field requires a valiant effort and I applaud you. The ephemeral nature of these cultures and the loss of cultures to Hurricane Harvey prevented the authors from conducting these experiments as thoroughly as they might want. The experimental design was unfortunately patchy, but there are important results here nonetheless. I have concerns about how the acetylene reduction assay was sampled and calculated. Unfortunately, the data provided did not answer them.

Validity of the findings

So many strains have been lost that there is no opportunity to replicate the experiments. The nitrogen addition experiments seem robust. If there are any DNA or RNA samples stashed away, they could be helpful in confirming some conclusions by determining whether the cultures were affected by viruses or whether host and asymbiotic strains had diverged.

I am trying to understand how the N2 fixation numbers were calculated and reported. It is a well-documented assay so I don't think the methods need to be extended, but perhaps an explanation to the editors how the data were handled might be helpful.

Additional comments

General comments:

It’s difficult to keep track of which species and strains were used in what experiment even though they are enumerated in the text. Please add a table listing the strains used and the measurements performed on each.

The culture isolation techniques described in the Results were very interesting, but I think they would be better served in the methods section. State what you learned, but leave the details in methods.

One confusing aspect of the paper is that Rhizosolenia is mentioned occasionally, but wasn’t used in the experiments. If there are no results to report, perhaps this paper should be written more clearly as a Hemiaulus paper.

Italicize “a” in chlorophyll a.

Specific comments:

Abstract:
- No hyphen is required for “nitrogen fixers” or “nitrogen fixation.” Do use a hyphen for the compound adjective “nitrogen-fixing,” however. Check the use of hyphens here and throughout.
- The phrase “a cyanobacteria-diatom symbiosis” sounds as if you only isolated one DDA but then you report two Hemiaulus species.
- The first time you report a species, spell out the entire genus, even if you’ve already used the genus for another species.
- Hypenate “symbiont-free.”

Introduction:

-Line 35: “Richelia” is misspelled.
-Line 56: Delete “cc” from “medium.”
-Line 64: This entire paragraph is about Hemiaulus, so it’s better if you don’t bring in new information about Rhizosolenia.

Methods and Materials:

Lines 86 - 87: Use plural for “strains” when you list multiple strains.

Lines 114 - 115: Were the cultures acclimated to the 2-L flasks or did you transfer them straight from the 50-mL tubes to the flasks?

Lines 125 - 126: It is not clear to me which light levels ere used in the spectrophotometer and which light levels were used to grow the cultures. Some of my comments elsewhere reflect that confusion. In addition, did you determine an optimal light level for each strain?

Line 139: Why the hyperbolic tangent? Is it the model that fit your data best? The hyperbolic tangent model is purely empirical with no theoretical biology supporting it. If it suits your data best, then use it. But if you haven’t tried one of the exponential functions, they can be derived from first principals. See Aalderink and Jovin, 1997, Journal of Plankton Research 19: 1713 - 1742.

Line 148: How many different start times did you use in a 24-hour period?

Line 155: “Nucleopore” is a brand. I presume these are polycarbonate? PC has a dye that extracts in methanol. Did you use a filter in your blank measurements?

Line 160: What kind of filter? What material was it made from?

Results:

Lines 174 - 175: Do you know what kinds of contaminants you had in your isolates? Were there phytoplankton other than the target diatom and symbiont cyanobacteria?

Lines 190 - 191: Did you measure cell height?

Line 206: “Both” light conditions? You listed 7 levels in your methods.

Lines 208 - 210: In the graph it seems that HL symbioses had more chl a than LL. This seems counter-intuitive to me when considering photoacclimation. What do you think is going on?

Line 209: Define “symbiosis” earlier at the first use.

Lines 219 - 220: Could another curve work?

Discussion:

Lines 258 - 261: This is a very interesting result and is in contrast to cyanobacteria, which fare better in natural oligotrophic seawater.

Lines 263 - 272: Did you take any samples for DNA or RNA analysis? Perhaps a colleague could help you look for evidence of viruses.

Lines 335 - 337: Do you have any genetic material preserved to determine whether these host and asymbiotic strains did diverge?

Lines 345 - 346: What environment light levels are relevant? Did you go high enough?

Figures:

Fig. 1: Excellent micrographs. If possible, please include a higher magnification to better see the structure of the cyanobacteria.

Fig 3: Legend, please! And you used 7 light levels. Why are only three shown?

Fig 4: This is an astonishing number of points. How many aliquots did you take throughout the day to add acetylene? What do the errorbars represent? Each point on the graph should represent multiple measurements in duplicate for one aliquot amended with acetylene. Multiply each point by 6 - 10, and it becomes an inhuman number of jabs in the GC. I’m not sure if the 5-point mean is meant to be a moving mean or something else. And the 4-h incubation is unclear to me.

Figure 6: What was fit to a hyperbolic tangent function? There’s no P-E curve here.

Reviewer 3 ·

Basic reporting

Overall the text reads fine. I have suggested a few revisions.

Citations and discussion are heavily focusing on a narrow list of authors. An example of a relevant early reference that could be additionally included and discussed:

John F. Heinbokel 1986 OCCURRENCE OF RICHELIA INTACELLULARIS (CYANOPHYTA)WITHIN THE DIATOMS HEMIAULUS HAUKII ADN H. MEMBRANACEUS OFF HAWAII J Phycol
https://doi.org/10.1111/j.1529-8817.1986.tb00043.x

The figures are ok. I have suggested a few edits to axes. One curve fit appears missing (see detail under General comments).

The manuscript is self-contained, but some additional detail should be provided. See below.

Experimental design

The manuscript reports important data on methods of isolation of a marine diazotroph symbiosis. There are not many known cultures of these associations, and very few laboratory observations.

I have some comments and suggestions regarding clarity of the data presentation. Several methods should be explained in more detail.

I suggest the authors include a table including key information about the specific strains they used in the experiments, to help reference these strains and experiments in any future studies.

Validity of the findings

Overall the data should be useful to the N2 fixation research community
Replication is not always clear. Was there biological replication in the experiments or are the means measurements from a single culture over time?
‘The validity of using a ‘5-point running average’ for determining N2 fixation rates is not possible to assess without sufficient detail about the AR assay setup (see comments below).
One or two anecdotal statements without sufficient evidence are included in the discussion and should be removed.

Additional comments

Title: The title reads a bit cumbersome and unclear. I suggest re-wording such as:
Isolation, growth, and nitrogen fixation rates of the oceanic Hemiaulus-Richelia (diatom-cyanobacterium) symbiosis in culture

Abstract:

The abstract should add a sentence to clarify whether Hemiaulus was cultured in a monoculture or whether the stated growth and N2 fixation rates were done while it was kept in the symbiosis with the host.
Diazotrophic symbiosis>reword. The symbiosis likely involves not just diazotrophy
Maximum growth rates of H. hauckii symbioses> remove the word ‘symbioses’
the H. membranaceus symbiosis> remove the words ‘the’ and ‘symbiosis’

Narrative:
L2 prokaryotic and eukaryotic
L4 replace ‘these oligotrophic seas’ by ‘in the open ocean’
L7 Here and elsewhere: check the journal requirements about ordering citations. Typically you would mention the earliest citation first (here Villareal 1992).
L55-56 The sentence about Calothrix is not well tied to the rest of the paragraph.
L64 this may also BE true for…
L70 revise the sentence for clarity/remove redundancy regarding ‘growth rate’ and ‘growth rate response’.
L72 ‘Modeling blooms’ is a bit ambiguous. Modeling bloom formation and fate?
L84 uM > um. The former is a unit for micromoles.
L87-90 State the purity grade of the chemicals.
L93 Unclear what you mean by ‘if required’.
L97 delete ‘quanta’ from the unit
L106 Remove the weblink. Replace with a citation to a permanent source for the information (such as an external data repository) or remove entirely.
L110 Unclear what you mean by ‘given experiment’. Specify which experiment or state more clearly.
L113 The wording “symbioses growth” is ambiguous as the symbiosis is composed of two organisms. State more specifically. Do you mean, growth of Hemiaulus and Richelia and N2 fixation of the host-symbiont association?
L114-115 Were there experimental replicates?
L115 Erlenmeyer
L116-117 Is there a reason why nitrate concentration was 4x the time that of ammonium?
L117 Was the salinity periodically checked? Evaporation is likely to have increased it over time.
L123-124 ‘Gradient’ here is misleading. Each treatment is a distinct light level, thus the experiment did not test the influence of a light gradient. The experiment tested the influence of distinct, constant light intensities.
L129 > “In addition, H. hauckii strain #92…”
L132 Cell counts were performed > Cells were counted
L140-141 Do you mean the N2 fixation curves were forced to zero? Why was this different from the growth curves?
L140 This term>The y-intercept term
L142 More information is needed on how the assay was conducted. What was your vial size, liquid volume, gas volume, gas injection volume? What acetylene was used? How long did you incubate each vial and under what conditions? How much gas was removed from the vial per time point? How many times was each vial sampled?
L144 ethylene conversion ratio 4:1> this needs to be stated more clearly. What does the 4:1 stand for?
L151 ‘center point of the time differential’> reword for clarity
L151 ‘5 point mean of the multiple time series...’ Are you referring to replicate vials of the same treatment? Reword for clarity.
L152-153 A citation to a MS thesis is not sufficient here to explain how the rates were calculated. Include the information in the methods and include a supplement if need be.
L153 How did you calibrate the GC? Did you run calibrations during the day?
L154 Do not underline a in Chlorophyll a – it should be italized. Abbreviate consistently through the text as chl a (or Chl a).
L158-159 Nutrient methods need to be explained with appropriate citations, including detection limits.
L170 what is sterile filtered seawater – explain how it was made
L175 >The use of these techniques resulted in…
L177 Delete ‘as well’
L179-180 Have been found by whom? Is this your personal experience? If so, revise sentence to indicate it.
L181 Laboratory structure? What is this? A partially covered wetlab building? A device placed on top of the bin?
L181-182 Unclear what ‘this site’ is referring to.
L190-191 Diameter? The cells are not spherical. Please state all the dimensions.
L204 time>times
L205 ‘chl per symbiosis’ is confusing. Do you mean per cell of Hemiaulus?
L208-209 the definition of chl a per symbiosis should be stated earlier. In the Figure 2 you use chl a per cell though. This is confusing.
L228 scotophase?
L230-231 photophase or photoperiod? Would help the reader to stick to consistent terminology.
L249-253 The information about the symbiont free Hemiaulus is anecdotal with no other evidence shown except a statement in the narrative. Are there any observations of this strain that could be shared such as morphology, growth rates etc. Without such data the paragraph should probably be removed.
L283 Website citation here needs to be changed into a citation to a permanent source or company name, city, state, country.
L285 What does ‘their rates’ mean? Did you convert their cell numbers to rates? Or did you take rates from the paper? If using actual rates from the paper it is unclear what the digitizing was used for.
L287 With ‘these rates’ are you referring to rates in Carpenter et al. 1999?
L299 nifH gene expression?
L306 >at least two mechanisms that..
L305-309 I suggest breaking this sentence into two to clarify the meaning.
L314 utilized > used
L316 utilize > use
L331 The statement about origin of the organisms is unclear. The origin of the cyanobacterium is likely earlier than the host.
L339 ‘symbioses blooms’ This wording is awkward > reword.

Fig. 1. These are beautiful images. What filters were used for the fluorescence images? Are these images overlays of images with two filter sets? State in the figure legend. Remove ‘image credit’ – no need to credit yourself.
Fig. 2. Delete ‘quanta’. Chlorophyll > Chlorophyll a.
B: Specify whether you are showing chla per Hemiaulus or chla per Richelia cell?
Fig. 3. The strain with x- symbols does not appear to have a Growth-I curve associated with it. The light affinity appears substantially different for this strain. This result does not seem to be discussed in the text as you only refer to two G-I curves and a very similar alpha for the two.
Fig.4. ‘average… calculated from the slope’ > reword to clarify. How long were the incubations for the black symbols?
Fig.5. ‘slope of successive measurements over a 4 h period’: This is confusing. Explain what you mean by a SLOPE of successive measurements. Are you referring to measurements representing a different cumulative incubation time from the same vial? If so, isn’t the comparison simply the rate/time from each of these measurements. This is not a slope.
Fig. 6. Should the y-axis be ‘% symbiotic Hemiaulus’? If not, the results do not correspond with the text narrative.

---

## Round 0.2 · Minor Revisions

I am looking forward for your point-by-point reply to these minor comments.

Reviewer 1 ·

Basic reporting

As mentioned in my prior read of this manuscript, the authors have written a clear, unambiguous, professional paper with sufficient literature background and context, figures, tables, and raw data. The manuscript is self-contained and after the minor revision, has met all the issues brought up previously.

Experimental design

The authors have done an excellent job of addressing all comments, adding clarity where needed, and describing statistical methods as necessary. Methods are very well written and will spark new investigations into these important associations.

Validity of the findings

As mentioned earlier, this manuscript is an important contribution to the state of knowledge for these keystone species.

Additional comments

The authors have addressed all of my noted issues in the revision.

Reviewer 2 ·

Basic reporting

Description of the experimental design, especially for N2 fixation measurements, is much more clear now. Thank you.

Specific comments:

L17: Hyphenate compound adjectives (“nitrogen-fixing”) but not nouns (“nitrogen fixation” or “nitrogen fixers”).

L55: Insert a space between “fixation” and “(.”

L56: Diazotrophy has been reported in the Arctic Ocean as well. See Harding et al, 2018, PNAS 115 (52) and other publications regarding UCYN-A.

L111: N2 abounds, so perhaps you should specify “fixed N.”

L122: Either use a comma on both sides of “where examined” or leave them out completely.

L140: Insert degree sign after “27” and a comma between lat and lon.

L156: I think rinsing implies discarding, so you don’t need to state it.

L252 - 266: Put the acetylene reduction assay description with the rest of the N2-fixation description.

L329 - 330: “Light-saturated growth” should be hyphenated while “light saturation” should not be.

L363: “Symbiont-free”

L366: Insert “microscopy.”

L393 - 395: Be consistent in using “fmol N2.”

Table 1: State the strains used.

Figure 2: Put the strain name in context somewhere in a sentence rather than a free-standing phrase. And capitalize “chlorophyll” after “B.”

Figure 3: Be more explicit about the circles. From the graph it looks as if they extend the experiment with the closed squares rather than the open squares.

Figure 4: State what the error bars represent.

Experimental design

With the clearer description of the methods, I no longer have reservations about the experimental design.

L164: That’s a good success rate! I applaud you.

L171 - 175: Any hypotheses on why YBCII worked better?

Validity of the findings

L319 - 321: I find it interesting that the chl per symbiosis was the same. Is photoacclimation not generally a trait for these diatoms?

L346 - 353: Is light or growth affecting N2 fixation rates? If you normalize N2 fixation rates to growth rates instead of volume or biomass, do you see a difference among the treatments?

L396 - 399: Maybe it’s light directly, but maybe all metabolic processes are ramped up. Again, I encourage you to normalize N2 fixation rates to growth rates to compare. Or do you have a proposed mechanism for how light directly affects N2 fixation?

L407 - 408. Goodness, yes!

Additional comments

I’m looking forward to seeing more DDA isolates in the future.

Reviewer 3 ·

Basic reporting

x

Experimental design

x

Validity of the findings

x

Additional comments

x

Annotated reviews are not available for download in order to protect the identity of reviewers who chose to remain anonymous.

---

## Round 0.3 · accepted · Accept

Thank you for addressing all points raised by the reviewers.